# Systemic Administration of Pegylated Arginase-1 Attenuates the Progression of Diabetic Retinopathy

**DOI:** 10.3390/cells11182890

**Published:** 2022-09-16

**Authors:** Ammar A. Abdelrahman, Katharine L. Bunch, Porsche V. Sandow, Paul N-M Cheng, Ruth B. Caldwell, R. William Caldwell

**Affiliations:** 1Department of Pharmacology and Toxicology, Medical College of Georgia, Augusta University, Augusta, GA 30912, USA; 2Culver Vision Discovery Institute, Medical College of Georgia, Augusta University, Augusta, GA 30912, USA; 3Bio-Cancer Treatment International, Bioinformatics Building, Hong Kong Science Park, Tai Po, Hong Kong SAR 511513, China; 4Vascular Biology Center, Medical College of Georgia, Augusta University, Augusta, GA 30912, USA; 5Department of Cell Biology and Anatomy, Medical College of Georgia, Augusta University, Augusta, GA 30912, USA

**Keywords:** diabetes, inflammation, oxidative stress, MΦ/microglia, arginase 1, retina

## Abstract

Diabetic retinopathy (DR) is a serious complication of diabetes that results from sustained hyperglycemia, hyperlipidemia, and oxidative stress. Under these conditions, inducible nitric oxide synthase (iNOS) expression is upregulated in the macrophages (MΦ) and microglia, resulting in increased production of reactive oxygen species (ROS) and inflammatory cytokines, which contribute to disease progression. Arginase 1 (Arg1) is a ureohydrolase that competes with iNOS for their common substrate, L-arginine. We hypothesized that the administration of a stable form of Arg1 would deplete L-arginine’s availability for iNOS, thus decreasing inflammation and oxidative stress in the retina. Using an obese Type 2 diabetic (T2DM) *db/db* mouse, this study characterized DR in this model and determined if systemic treatment with pegylated Arg1 (PEG-Arg1) altered the progression of DR. PEG-Arg1 treatment of *db/db* mice thrice weekly for two weeks improved visual function compared with untreated *db/db* controls. Retinal expression of inflammatory factors (iNOS, IL-1β, TNF-α, IL-6) was significantly increased in the untreated *db/db* mice compared with the lean littermate controls. The increased retinal inflammatory and oxidative stress markers in *db/db* mice were suppressed with PEG-Arg1 treatment. Additionally, PEG-Arg1 treatment restored the blood–retinal barrier (BRB) function, as evidenced by the decreased tissue albumin extravasation and an improved endothelial ZO-1 tight junction integrity compared with untreated *db/db* mice.

## 1. Introduction

Diabetic retinopathy (DR) is a common and potentially blinding microvascular complication of diabetes caused by prolonged hyperglycemia, hyperlipidemia, and oxidative stress [1,2]. DR affects many Type 1 and Type 2 diabetic (T2DM) patients and is characterized by inflammation, ROS formation, and hypoxia. The inflammatory state is driven by increased production of pro-inflammatory cytokines and factors, which leads to alterations in the retinal blood flow, relative hypoxia, and upregulation of vascular endothelial growth factor (VEGF). This environment leads to breakdown of the blood–retinal barrier (BRB), cytotoxicity, and neurovascular degeneration [3].

Current DR therapeutic modalities target advanced stages of the disease. Laser pan-retinal photocoagulation (PRP) has been used for the treatment of DR, but has substantial risk, destroys the peripheral retina, and its technical requirements limit patient accessibility [4]. Anti-VEGF agents are one of the few classes of drugs approved for the treatment of diabetic macular edema and proliferative DR [5]. These agents are delivered by intravitreal injection with variable success and are associated with poor patient compliance, transiently increased intraocular pressure, and an increased risk of endophthalmitis, an ocular emergency. Moreover, there is concern about off-target effects, including adverse cardiovascular changes and disruption of retinal neuron homeostasis, given that VEGF has a critical role in the physiologic adaptation of neurons [6,7]. Thus, novel strategies to prevent, halt, or reverse DR progression are greatly needed [8].

Prior studies by us and others on diabetes and oxidative stress have found increased expression of the urea cycle enzyme, arginase 1 (Arg1), which hydrolyzes L-arginine into urea and L-ornithine [9,10,11,12]. Arg1 competes with nitric oxide synthase (NOS) isoforms, such as endothelial (eNOS) and inducible (iNOS), for their common substrate, L-arginine. In the vascular endothelium, the diabetic environment promotes increased Arg1 expression, which reduces L-arginine’s bioavailability for NO production by eNOS, resulting in vascular constriction, thrombogenesis, and leukostasis [12].

In contrast to its detrimental effects on vasculature, Arg1 has been shown to be neuroprotective. In DR, the inflammasome is activated, which results in the production of inflammatory cytokines. These cytokines and other inflammatory factors promote the activation and polarization of resident retina microglia and macrophages (MΦ) to the M1-like pro-inflammatory phenotype. These activated MΦ/microglia produce cytotoxic levels of NO via iNOS, which result in oxidative stress and neurovascular injury [11]. It has been demonstrated that elevated levels of Arg1 in these immune cells decrease the bioavailability of L-arginine, thus reducing both iNOS function and expression, resulting in suppression of the pro-inflammatory M1-like phenotype [13,14]. Additionally, elevated Arg1 increases the production of polyamines (putrescine, spermine, and spermidine) via the arginase/ornithine decarboxylase (ODC) pathway. These polyamines promote MΦ/microglia polarization from the M1-like pro-inflammatory phenotype to the reparative, anti-inflammatory (M2-like) phenotype with injury resolution via efferocytosis (clearance of cellular debris) [15,16,17].

Our group has recently demonstrated that Arg1 has a neuroprotective role in mouse models of both retinal ischemia/reperfusion injury and ischemic stroke, and further demonstrated the beneficial effects of Arg1 treatment in the form of intravitreal and intraperitoneal injections [18,19]. Recombinant Arg1 covalently linked to polyethylene glycol (PEG-Arg1) possesses a significantly prolonged half-life compared with the native enzyme. Administration of PEG-Arg1 in both mouse models may protect against neurodegeneration by promoting an anti-inflammatory, reparative M2-like MΦ/microglia phenotype.

This study examined the effectiveness of PEG-Arg1 administration (via intraperitoneal delivery) in DR progression using the obese/diabetic *db/db* mouse model. The *db/db* mouse has been reported to exhibit DR that closely resembles the early phases of human DR and shows T2DM metabolic aberrations that do not occur in Western-diet-fed wild-type mice with comparable obesity [20,21].

## 2. Materials and Methods

### 2.1. Animal Model

All the animal studies performed followed the Association for Research in Vision and Ophthalmology (ARVO) Statement for the use of animals in ophthalmic and vision research and were approved by the Augusta University Institutional Animal Care and Use Committee. All animals were maintained at an ambient temperature with a 12:12 h light/dark cycle and were fed ad libitum. C57BL/KsJ-*db/db* (Stock No. 000642) female mice were obtained from Jackson Laboratories (Bar Harbor, ME), and two separate studies were conducted. The first study characterized the retinas of the obese diabetic *db/db* mice and compared them with their lean heterozygote Db/+ littermate controls between 4 and 6 months of age. In the second study, *db/db* mice between 4 and 6 months of age were treated with intraperitoneal (IP) injections with either a 25 mg/kg dose of PEG-Arg1 or the same dose and volume of PEG-5000 MW (as a vehicle control) thrice weekly for 2 weeks. Body weight was measured before each injection to ensure adequate dosing and to monitor weight changes. Random blood glucose levels collected from tail tip samples were measured using a glucometer on Days 0, 7, and 17 from the start of treatment. Mice were sacrificed 3 days after the last dose, and their whole eyes were collected and prepared for analysis.

The pharmaceutical grade PEG-Arg1 used for these studies was provided as a kind gift by Bio-Cancer Treatment International Limited (BCT, Hong Kong). PEG-Arg1 is recombinant human arginase 1 covalently attached to methoxy polyethylene glycol (mPEG-SPA; MW 5000) via succinamide propionic acid (SPA) linkers to increase the stability and in vivo half-life (the estimated half-life is ~3 days versus a few minutes for the native enzyme) [22]. The product of the pegylation process has 1 to 6 PEG units attached to the lysine residues of rhArg1; a detailed chemical characterization of PEG-Arg1 was described by Tsui et al. [23].

### 2.2. Visual Function Studies

Visual acuity and contrast sensitivity in mice were assessed using optokinetic response tracking (OKT) (Cerebral Mechanics, Inc., Lethbridge, AB, Canada) as previously described [24]. Briefly, sine wave gratings were displayed across four LCD screens revolving around a central stand. The optomotor reflex of an unrestrained mouse to the rotating vertical sine wave grating was recorded by manual tracking of reflexive head movements. To determine the spatial frequency thresholds, an increasing stairstep algorithm was utilized, starting at 0.042 cycles/deg (*c/d*) with 100% contrast. To determine the contrast threshold, the spatial frequency was fixed at 0.092 *c/d* and a decreasing stairstep algorithm was utilized, starting with the maximum contrast. Manual tracking of the visual reflexes of the mice was performed by an investigator blinded to their treatment status. Readings from clockwise (for the left eye) or counterclockwise (for the right eye) were recorded and the average for each mouse was used in the data analysis.

### 2.3. Electroretinogram (ERG) Studies

The electroretinographic waveforms in response to photopic and scotopic light were assessed in both *db/db* and lean control mice using the Celeris Fully Integrated ERG Testing system (Diagnosys LLC, Lowell, MA, USA). Mice were dark-adapted overnight and anesthetized using intramuscular ketamine and xylazine. Tropicamide (0.5%, Akorn, Lake Forest, IL, USA) and phenylephrine HCl (2.5%, Paragon, Portland, OR, USA) ophthalmic drops were applied to induce mydriasis. LED probes were used to generate light stimuli and scotopic light responses at increasing intensities (0.001, 0.005, 0.01, 0.1, 0.5, 1 cd s/m^2^), and the responses were recorded. Subsequently, the mice were light-adapted and their photopic light responses to increasing stimulus intensities (3, 10, 25, 50, 100, 150 cd s/m^2^) were measured. Readings from both the right and left eyes were recorded, and the average for each mouse was used in the data analyses.

### 2.4. Tissue Collection and Preparation

Mice were anesthetized with isoflurane and euthanized via exsanguination. Retinal vascular blood was removed by perfusion with cold phosphate buffer saline (PBS) through the left ventricle for 5 min. From each mouse, one retina was flash-frozen in liquid nitrogen and stored at −80 °C until homogenized in RIPA buffer with protease and phosphatase inhibitors for Western blot analyses. The other eye was enucleated and fixed in 4% paraformaldehyde (PFA) overnight at 4 °C, followed by incubation, first in a 15% and then in a 30% (*w*/*v*) sucrose solution. Fixed eyes were embedded in an optimal cutting temperature (OCT) compound at −80 °C for histologic sectioning. In select mice, the eyes were fixed in 4% PFA at room temperature overnight, and then dissected to extract the neural retinal cup for retina flat-mount preparation. Radial, symmetric cuts were made to the retinal cup to create a four-petal flower shape, which is optimal for the application of a coverslip after immunohistochemical labeling.

### 2.5. Immunofluorescence Staining and Visualization

Eyes from each experimental group were processed as previously described for sectioning and immunofluorescent labeling [24]. The following primary antibodies were used: 3-nitrotyrosine (3-NT) (Sigma-Aldrich Cat. No. N0409, St. Louis, MO, USA; 1:300), 4-hydroxynonenal (4HNE) (Abcam Cat. No. ab46545, Cambridge, MA, USA; 1:50), zonula occludens-1 (ZO-1) (Invitrogen, Cat. No. 61-7300, Waltham, MA, USA; 1:50), platelet endothelial cell adhesion molecule-1 (PECAM-1/CD31) (BD Biosciences, Cat. No. 550300, Franklin Lakes, NJ, USA; 1:50), ionized calcium-binding adaptor molecule 1 (Iba-1) (Wako Chemicals USA Inc., Cat No. 019-19741, Richmond, VA, USA, 1:500), biotinylated isolectin B4 (Vector Laboratories, Cat. No. B-1205, Burlingame, CA, USA; 1:20), and polyethylene glycol (PEG) (RevMab Biosciences, Cat. No. 31-1008-00, South San Francisco, CA, USA; 1:50). Imaging was performed using a Carl Zeiss 780 multiphoton confocal microscope.

To prepare retina flat-mounts for immunofluorescent labeling, the flat-mounts were first permeabilized using 1% Triton-X-100 in 1X PBS (PBS-1%TX). The flat-mounts were subsequently blocked in PBS-0.1%TX with filtered 10% normal goat serum and 1% bovine serum albumin (BSA) for 1 h at room temperature while rocking gently. Retinas were then incubated overnight at 4 °C with one of the primary antibodies. After the primary antibody has been removed, the retinas were washed 3 times for 10 min in 1× PBS and incubated in a secondary antibody either conjugated to Alexa Fluor 647 (far red), Alexa Fluor 594 (red), or Texas Red Avidin D (Vector Laboratories, Cat. No. A-2006-5, Burlingame, CA, USA) for 1 h at room temperature. The flat-mounts were then washed 3 times for 10 min in PBS-0.1%TX and mounted with coverslips using SlowFade Gold Antifade Mountant (Invitrogen, Cat. No. S36936, Waltham, MA, USA). The flat-mounts were imaged using a Leica Stellaris 5 confocal microscope.

### 2.6. Assessment of Macrophage/Microglia (MΦ/Microglia Soma Sizes)

Retinal flat-mounts labeled for both isolectin B4 and Iba-1 (as described above) were imaged as Z-stacks using a confocal microscope, and the images were processed using LAS X software [19]. Image acquisition and processing were carried out by an investigator blinded to the groups. Exported images of Iba-1^+^ MΦ/microglia were analyzed using the automated methods developed by Davis et al. [25]. Automated calculation of the mean Iba-1+ MΦ/microglia soma size was performed using Fiji-ImageJ2 software by applying pre-recorded ImageJ macrocodes [25]. An average of 8 images from each retina was used for statistical analysis.

### 2.7. Western Blot Analysis

Retinal protein lysates were separated on sodium dodecyl sulfate (SDS) polyacrylamide gels and transferred to polyvinylidene fluoride (PVDF), where they were blocked in 3% BSA (Bio-Rad, Hercules, CA, USA) and then incubated overnight at 4 °C with one of the following primary antibodies prepared in 3% BSA: IL-1β (R & D Systems, Cat. No. AF-401-NA, Minneapolis, MN, USA; 1:1000), tubulin (Sigma-Aldrich Cat. No. T-9026, St. Louis, MO, USA; 1:1000), heat shock protein 90 (Hsp90) (BD Biosciences, Cat. No. 610418, Franklin Lakes, NJ, USA; 1:1000), albumin (Proteintech Cat. No. 16475-1-AP, Rosemont, IL, USA; 1:1000), or TNFα (Abcam, Cat. # ab1793, Cambridge, MA, USA, 1:1000). The following day, the membranes were washed 3 times in Tris-buffered saline with 0.5% Tween-20 (TBS-T) and then incubated with the corresponding horseradish peroxidase-conjugated secondary antibody (GE Healthcare, Piscataway, NJ, USA; 1:1000) for 1 h at room temperature. Signals were detected using an enhanced chemiluminescence system (GE Healthcare Bio-Science Corp., Piscataway, NJ, USA) and quantified by densitometry using ImageJ software (version 1.49, National Institutes of Health, Bethesda, MD, USA) and normalized to the loading control.

### 2.8. Statistical Analysis

Data are presented as the mean ± SEM. Statistical analyses were performed using Student’s t-test or analysis of variance with Tukey’s post-test. Values of *p* < 0.05 were considered statistically significant. These analyses were performed using GraphPad Prism, version 4.00 (GraphPAD Software Inc., San Diego, CA, USA).

## 3. Results

### 3.1. Retinal and Visual Dysfunctions in db/db Mice

In our initial study, we compared the electroretinography (ERG) and optokinetic (OKN) responses and of obese diabetic female *db/db* mice with those of age-matched lean DB/+ controls. In scotopic conditions, the ERG a-wave amplitude of *db/db* mice was lower than that of the lean controls (Figure 1A). The ERG b-wave amplitude of the *db/db* mice also was significantly lower than that of the controls (Figure 1B), as previously shown in male *db/db* mice [20]. Similarly, the amplitudes of the ERG b-wave of the photopic responses were significantly lower in the *db/db* group than in the DB/+ controls. However, the ERG photopic a-waves’ amplitude did not differ significantly between the two groups (Figure 1C,D). Optokinetic responses demonstrated significantly worse visual acuity and contrast sensitivity in *db/db* mice compared with the lean DB/+ controls (Figure 1E,F).

### 3.2. Systemic PEG-Arg1 Treatment Restores Visual Function in db/db Mice

Our prior studies demonstrated the neuroprotective effect of treatment with PEG-Arg1 in murine models of retinal ischemia/reperfusion injury, optic nerve crushing, and ischemic stroke [18,19]. We have also shown that systemic PEG-Arg1 treatment, through IP injection, penetrates the blood–brain-barrier (BBB) and the blood–retinal barrier (BRB) under conditions of ischemia and injury [19]. In this study, we examined the effect on the visual function of PEG-Arg1 treatment (compared with the PEG control) in *db/db* mice for 2 weeks. We observed that the PEG-Arg1-treated *db/db* mice had significantly improved visual acuity and contrast sensitivity compared with *db/db* mice which had received only the PEG control (Figure 2A,B).

It is noteworthy that the PEG-Arg1 treatment did not affect body weight or random blood glucose levels of the *db/db* mice compared with their PEG control-treated counterparts (Figure 2C,D). The pharmacokinetic/pharmacodynamics of PEG-Arg1 have been extensively studied in mice and humans [23,26]. We have previously shown the bioavailability of PEG-Arg1 in the retina after exposure to ischemia/reperfusion injury due to breakdown of the blood–retinal barrier [19]. Using Western blot analysis and immunofluorescence labeling (Appendix A) for PEG in frozen retinal sections, we confirmed the bioavailability of PEG-Arg1 (>250 kD) in the neural retina of *db/db* mice 3 days after the last IP injection.

### 3.3. Elevated iNOS Expression and Oxidative/Nitrative Stress in db/db Retinas

Elevated oxidative/nitrative stress is a hallmark of DR that serves to drive disease progression [27,28]. Western blot analyses of retinal protein lysates showed that *db/db* retinas had significantly higher iNOS expression than the retinas of nondiabetic lean DB/+ control mice (Figure 3A,B). The *db/db* mice exhibited higher retinal levels of 4-hydroxynonenal (4-HNE), a product of lipid peroxidation (Figure 3C,D). Additionally, immunoreactivity to 3-nitrotyrosine (3-NT), a marker of peroxynitrite (a potent oxidant), was higher in *db/db* retina cross-sections compared with the DB/+ controls (Figure 3E,F).

### 3.4. PEG-Arg1 Treatment Decreases iNOS Expression and Oxidative/Nitrative Stress in db/db Retinas

Western blot analyses were used to determine the effects of systemic PEG-Arg1 treatment on retinal iNOS expression. The expression of iNOS was found to be significantly reduced in the retinas of PEG-Arg1-treated *db/db* mice compared with the PEG-treated controls (Figure 4A,B). In frozen retina sections, lipid peroxidation was quantified by the 4-HNE immunofluorescence intensity. Levels of 4-HNE were significantly lower in PEG-Arg1-treated *db/db* retinas compared with the PEG-treated *db/db* controls (Figure 4C,D). PEG-Arg1 treatment also significantly reduced the levels of 3-NT in *db/db* retina sections compared with the PEG-treated controls (Figure 4E,F).

### 3.5. Increased Retinal Inflammation in db/db Mice

Pro-inflammatory cytokines, such as IL-1β and TNF-α, are elevated in the retinas of both diabetic rodents and humans [29,30,31]. We investigated the inflammatory response in the retinas of *db/db* mice via Western blot analyses and immunofluorescent labeling. The Western blot analyses of *db/db* retinal protein lysates showed a significantly higher expression of IL-1β than in lean DB/+ controls (Figure 5A,B). To determine the potential role of MΦ/microglial cell activation in the inflammatory response, retina flat-mounts were labeled for Iba1 (a nonspecific marker of MΦ and microglia) and isolectin B4 (a vascular marker). Iba1 and isolectin B4 co-labeling allowed for an analysis of MΦ/microglia morphology, which has been strongly correlated with the MΦ/microglia phenotype [32]. Morphometric analyses of MΦ/microglia were performed to assess soma size in the superficial vascular plexus of the retina. Analyses showed that the soma of MΦ/microglia in *db/db* retinas were significantly larger than in the lean DB/+ controls (Figure 5C,D). The enlarged soma size and amoeboid morphology are consistent with the M1-like pro-inflammatory activated phenotype, suggesting that activated MΦ/microglia contribute to inflammation, ROS production, and subsequent BRB breakdown in *db/db* retinas [25,33].

### 3.6. PEG-Arg1 Reduced the Retinal Expression of Pro-Inflammatory Cytokines in db/db Mice

Prior experiments carried out in murine models of retinal ischemia/reperfusion injury and optic nerve crushing showed that PEG-Arg1 treatment inhibited the activation of pro-inflammatory MΦ/microglia and reduced retinal cytokine levels [19]. To determine the effects of PEG-Arg1 treatment on retinal inflammation in *db/db* mice, Western blot analyses of retinal protein lysates were performed. PEG-Arg1 treatment in *db/db* mice resulted in lower levels of the pro-inflammatory cytokines IL-1β and TNF-α compared with the PEG-treated controls (Figure 6A–D). Furthermore, morphometric analyses of the Iba1+ cells in the superficial vascular plexus of the retina showed that treatment of *db/db* mice with PEG-Arg1 significantly reduced the average soma size of the retinal MΦ/microglia (Figure 6E,F). Together, these results demonstrate the anti-inflammatory effects of PEG-Arg1 treatment on the retinas of the obese T2DM *db/db* mice.

### 3.7. PEG-Arg1 Treatment Restores the Blood–Retinal Barrier (BRB) in db/db Mice

Zonula occludens 1 (ZO-1) is a scaffold protein required for the assembly and function of the tight junction protein complex in many cell types, including vascular endothelial cells [34]. In diabetic mice, ZO-1 expression has been found to be markedly decreased, leading to breakdown of vascular endothelial cell tight junctions and reduced barrier function [35,36]. To assess the expression of ZO-1 in endothelial cells, retina sections were co-labeled with ZO-1 and cluster of differentiation 31 (CD31), an endothelial cell-specific marker. In PEG-treated control *db/db* mice, ZO-1 immunofluorescent labeling exhibited noncontiguous expression, with a higher number of breaks along the vasculature compared with PEG-treated lean DB/+ control mice. In contrast, PEG-Arg1-treated *db/db* mice had greater continuity of ZO-1 expression along the vessels, with fewer tight junction breaks (Figure 7A,B). To further assess BRB status, the level of albumin extravasation from the vasculature into the retinal tissue was determined via Western blot analysis. Retinal protein lysates obtained from mice perfused transcardially with a saline buffer were examined for the level of albumin. PEG-Arg1-treated *db/db* mice exhibited reduced albumin leakage compared with the PEG-treated control *db/db* mice, indicating improved BRB function with PEG-Arg1 treatment (Figure 7C,D). In conclusion, PEG-Arg1 treatment restored BRB function in *db/db* mice.

## 4. Discussion

This study expanded the characterization of the pathological features of DR in the context of the relationship of Arg1 and iNOS in a leptin receptor-deficient (*db/db*) T2DM mouse model. Further, the efficacy of systemic administration of PEG-Arg1 for the treatment of DR in this obese diabetic model was evaluated and was shown to result in improvements in visual function, restoration of BRB integrity, reductions in oxidative/nitrative stress and inflammation, and suppression of MΦ/microglia activation.

Due to the homozygous leptin receptor deletion, *db/db* mice do not receive the satiety signal in the brain and become hyperphagic, resulting in the development of obesity and T2DM by 8 weeks of age [37]. In addition to the development of insulin resistance causing chronic hyperglycemia, these mice also exhibit metabolic aberrations, including hypercholesterolemia, hypertriglyceridemia, and elevated free fatty acids, all of which mirror the macro- and microvascular complications seen in human T2DM patients [38]. The neurodegenerative aspects of human DR are recapitulated in the *db/db* mouse model and have previously been well characterized in male mice [20,39,40]. As early as 3 months of age, male *db/db* mice exhibit retinal thinning, decreased visual acuity, decreased scotopic a- and b-wave amplitude responses, and increased evidence of apoptosis in the inner retinal layers [20,39,40]. However, it has been recently reported that the prevalence of DR in T2DM patients is higher in females than in males [41]. To our knowledge, this study is the first to examine the visual deficits as well as a potential novel therapy for DR in female *db/db* mice. Our study revealed that female *db/db* mice exhibit retinopathy and declining visual function similar to their male counterparts (Figure 1A–F). Both visual acuity and contrast sensitivity were impaired in the female *db/db* mice. Both deficits were significantly improved by the PEG-Arg1 treatment.

As previously stated, prior studies have demonstrated neuroprotective effects of systemic and intravitreal treatment with PEG-Arg1 in murine models of ischemia/reperfusion injury and optic nerve crushing injury [18,19]. The protective effects were associated with reductions in the expression of iNOS and inflammatory cytokines, along with the suppression of MΦ/microglia activation. Despite having an ischemic component, the pathophysiology of DR is more complex, involving several other pathological factors that arise from metabolic dysfunction, including hyperglycemia, hypercholesterolemia, and increased free fatty acids.

In the healthy retina, resident MΦ and microglia are important components of retinal immune surveillance, the response to physiologic stressors, and normal homeostatic functioning of the retinal neurons [42]. Under normal conditions, activation of MΦ/microglia is a carefully regulated process, as the activated M1-like pro-inflammatory phenotype is neurotoxic [42,43]. However, dysregulation of MΦ/microglia activation results in prolonged and sustained inflammation, oxidative stress, neurodegeneration, glial dysfunction, and cell death [42]. In DR, MΦ/microglia activation is prevalent, contributing to both the progression and severity of the disease [32]. MΦ/microglia contribute to diabetic microvascular complications, including the development of DR, through elevated production of inflammatory cytokines, increased oxidative/nitrative stress, and impaired phagocytic functions [44,45,46,47,48,49]. The state of increased metabolic flux caused by hyperglycemia, hypertriglyceridemia, and elevated free fatty acids promotes an exaggerated MΦ/microglia inflammatory response and inhibition of the pro-resolution functions, including efferocytosis [48]. The role of pro-inflammatory cytokines, such as IL-1β and TNF-α, in the mediation of chronic inflammation and disruption of the BRB has been well documented in several animal models of diabetes and DR [29]. In our study, female *db/db* mice exhibited increased levels of pro-inflammatory cytokines in retinal protein lysates compared with their age-matched lean DB/+ littermates. Additionally, markers of oxidative/nitrative stress in the retinal sections of *db/db* mice were significantly elevated compared with the DB/+ controls. These findings of increased inflammation and damage in the context of impaired visual acuity are consistent with the development of DR.

Previous studies using diabetic human retinas have demonstrated that activated and hypertrophied MΦ/microglia cluster around retinal vessels that exhibit BRB dysfunction, neovascularization, and/or neurodegeneration [19,32]. In our study, exceptionally large Iba1-positive MΦ/microglia were found near larger blood vessels in the superficial vascular plexus (Figure 5F). The average soma size of these cells was significantly greater in the *db/db* retinas compared with the lean DB/+ controls, suggesting their activation to the pro-inflammatory M1-like phenotype [25]. Furthermore, the PEG-Arg1 treatment significantly inhibited this increase in soma size of the Iba1-positive perivascular MΦ/microglia, suggesting a reversal of their phenotype from the pro-inflammatory phenotype to the reparative phenotype.

Studies in the brain and retina have identified a specific subset of resident MΦ, the perivascular MΦ (PVMs) [50]. These cells are in contact with the abluminal surface of blood vessels or are located within one cell thickness from it, and function as sentinels to continuously sample and clear exogenous macromolecules from the neuronal microenvironment [51,52]. These cells constitutively express scavenger receptors as part of their immune function and have an important role in the uptake and accumulation of oxidized lipids and lipofuscin during the aging process [51,53]. The hyperglycemic, hyperlipidemic, and insulin-resistant conditions in the *db/db* retinas may contribute to the activation of such cells and decrease their ability to eliminate waste and apoptotic cell debris from the neuronal microenvironment, further perpetuating retinal dysfunction [45,54,55]. Additional studies are needed to explore this possibility.

Further study is also required to elucidate the specific mechanisms responsible for the beneficial effects of PEG-Arg1 in slowing or stopping the progression of DR. However, one likely explanation relates to its impact on iNOS expression and activity. The role of elevated iNOS and cytotoxic levels of NO in mediating diabetic retinal pathophysiology has been widely investigated [56]. Reductions in iNOS levels and/or its pharmacological inhibition have been shown to prevent or reduce the development of the early stages of DR [57,58]. The expression of iNOS is controlled by its substrate availability, and L-arginine depletion has been shown to reduce iNOS expression [14]. Additionally, the beneficial effects of L-arginine depletion have been reported in a study of acute spinal cord injury. The investigators found that the neuroinflammation associated with injury was attenuated through depletion of L-arginine via suppression of iNOS production in immune cells [59]. In addition to suppressing iNOS, the administration of PEG-Arg1 results in the increased production of downstream products, including polyamines (putrescine, spermine, and spermidine) [11]. Polyamines have been shown to reverse MΦ/microglia polarization from the M1-like pro-inflammatory phenotype to the M2-like anti-inflammatory phenotype [13,15,16]. We postulate that this dual effect is responsible for the efficacy of PEG-Arg1 treatment in *db/db* mouse DR.

Another likely mechanism for the beneficial effects of PEG-Arg1 treatment on DR involves an inhibitory effect on MΦ/microglia activation. For decades, Arg1 has been used as a marker of M2-like reparative MΦ/microglia, and its promotion of an anti-inflammatory phenotype has long been recognized [60]. While it has been well established that the effect of systemic PEG-Arg1 is primarily due to its depletion of circulating levels of L-arginine in cancers [61,62], the downstream mechanisms that underlie the anti-inflammatory and neuroprotective effects have yet to be fully elucidated. One possible mechanism is that due to the decreased intracellular L-arginine from the PEG-Arg1 treatment, the mechanistic target of rapamycin (mTORC) pathway is inhibited. Studies have shown that depletion of L-arginine from specific cell compartments can limit the activation of mTORC, a master regulator pathway that normally serves to modulate growth and the response to various stressors [63]. Inhibitors of mTORC have been shown to be effective in reducing excessive immune responses in cardiovascular disease and diabetes [64].

To our knowledge, this study reports, for the first time, the anti-inflammatory benefits of L-arginine depletion achieved by using PEG-Arg1 in the context of diabetes-associated neuroinflammation. Extrapolation of the experimental results to Type 1 diabetic conditions would require further studies in models of Type 1 diabetes. In addition to the lack of leptin receptor signaling in *db/db* mice in preventing appetite satiety, its absence could be a confounding factor contributing to and accelerating retinal degeneration, given the other effects of leptin signaling in the retina [24,65]

## 5. Conclusions

The growing number of diabetic patients with limited access to the gold standard therapies for the treatment of diabetic complications prompts the search for novel, more accessible treatments. Our results indicate that systemic L-arginine depletion using PEG-Arg1 is effective in reducing inflammation and oxidative stress, restoring BRB integrity, and improving visual acuity in a murine model of diabetic retinopathy. We have demonstrated that systemically delivered PEG-Arg1 can cross the BRB into the retina, especially in areas with pathological barrier dysfunction. Therefore, PEG-Arg1 could be used without the need for intravitreal delivery, removing the requirement for highly trained personnel to administer the treatment and reducing the risk of damage to the eye. In addition, PEG-Arg1 therapy has been well-tolerated in cancer patients, making PEG-Arg1 a promising therapeutic agent for DR, worthy of further investigation and development.

## Figures and Tables

**Figure 1 cells-11-02890-f001:**
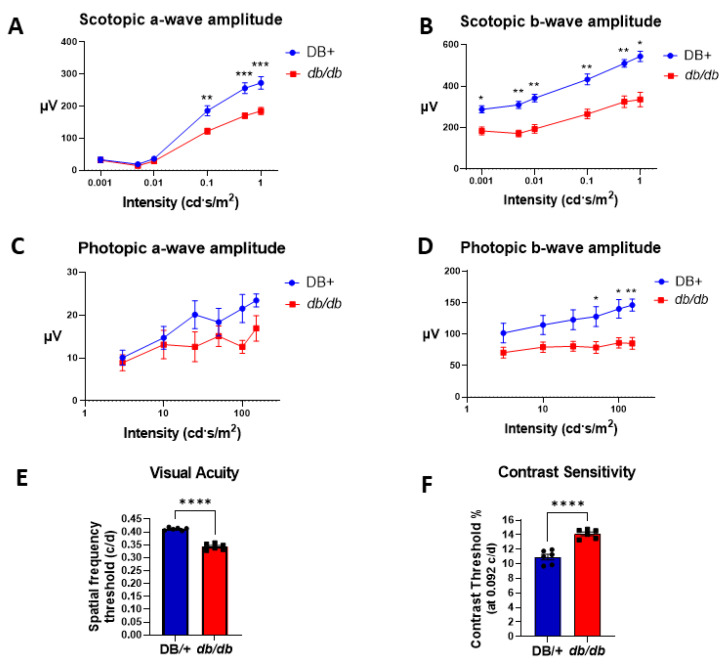
The *db/db* mice showed impaired retinal and visual function. (**A**,**B**) Scotopic electroretinography (ERG) responses and (**C**,**D**) photopic ERG responses were recorded in female *db/db* mice and lean DB/+ controls. Mice were dark-adapted overnight prior to the scotopic ERG response assessment. For photopic measurements, mice were exposed to a bright probe light for 5 min to saturate rod signals. Mean amplitudes of the scotopic and photopic a- and b-wave ERG components (**A**,**C** and **B**,**D**, respectively) with the SEM of flash responses are plotted against the luminance intensity (cd⋅s/m^2^). (**E**) The spatial frequency threshold (a measure of visual acuity) was significantly reduced in *db/db* mice compared with the lean controls. (**F**) The contrast threshold (the measure of minimum level of contrast that could be discerned by the mouse) was significantly impaired in the *db/db* mice compared with the lean controls. Data are presented as the mean ± SEM. * *p*< 0.05, ** *p*< 0.01, *** *p* < 0.001, **** *p <* 0.0001.

**Figure 2 cells-11-02890-f002:**
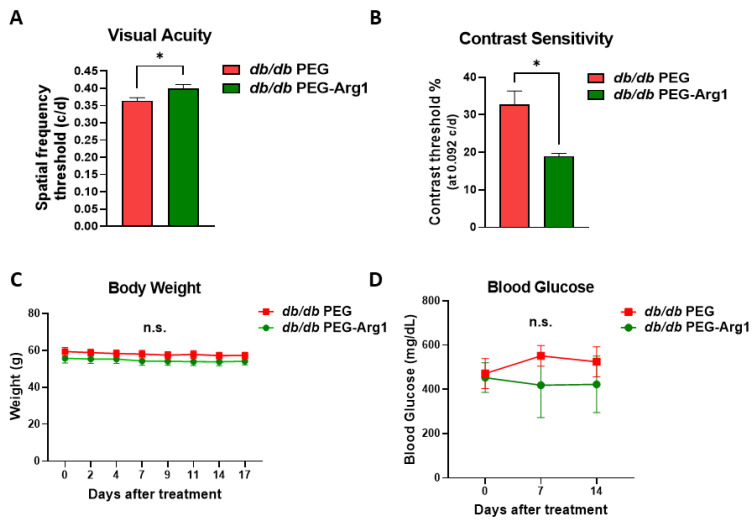
PEG-Arg1 treatment improves the visual function of *db/db* mice without affecting body weight or random blood glucose. (**A**,**B**) The *db/db* mice that received the PEG-Arg1 treatment showed improved visual acuity and contrast sensitivity compared with the PEG-treated *db/db* mice (n = 3 per group with * *p* < 0.05). (**C**) No significant differences appeared in the body weight of *db/db* mice treated with 25 mg/kg/dose of either PEG-5000 (control) or PEG-Arg1 thrice weekly for 2 weeks. (**D**) Randomly collected blood glucose levels showed no significant change with PEG-Arg1 treatment. Data are presented as the mean ± SEM.

**Figure 3 cells-11-02890-f003:**
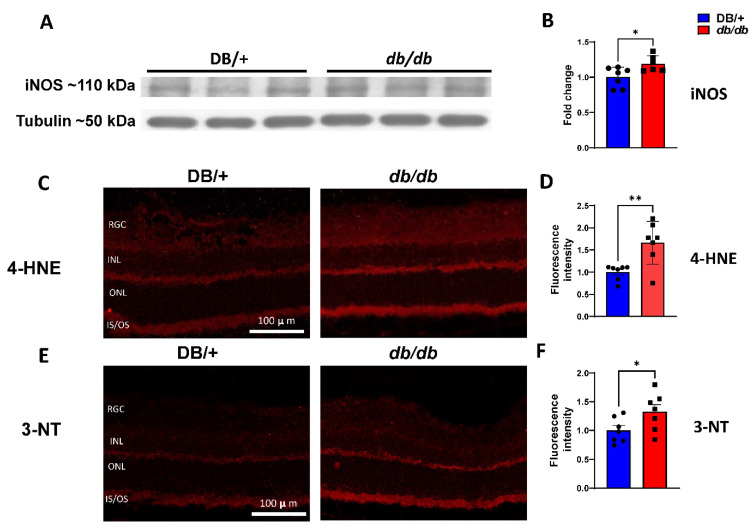
Increased oxidative/nitrative stress in the retinas of *db/db* mice. (**A**) Representative Western blots with (**B**) quantification expressed as fold changes from the control, showing elevated iNOS levels in *db/db* retinal protein lysate compared with the lean DB/+ controls (n = 6–7/genotype) (* *p* < 0.05). (**C**,**E**) Immunofluorescent labeling at 10× of frozen retina sections with 4-hydroxynonenal (4-HNE) and 3-nitrotyrosine (3-NT). (**D**,**F**) Fluorescence intensity quantification of 4-HNE and 3-NT, respectively (n = 7/genotype). Data are presented as the mean ± SEM. (* *p* < 0.05, ** *p* < 0.01). RGC: retinal ganglion cell layer; INL: inner nuclear layer; ONL: outer nuclear layer; IS/OS: inner/outer segment of photoreceptor.

**Figure 4 cells-11-02890-f004:**
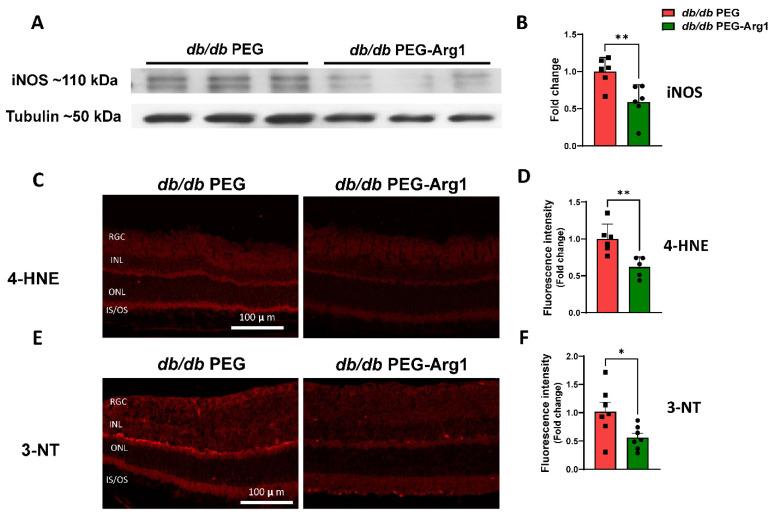
PEG-Arg1 treatment reduces oxidative/nitrative stress in *db/db* retinas. (**A**) Representative Western blots of retinal iNOS levels with (**B**) quantification, demonstrating significant reductions in iNOS protein expression in PEG-Arg1-treated *db/db* mice compared with the PEG-treated controls (n = 6/group, ** *p* < 0.01). (**C**,**E**) Representative immunofluorescent labeling at 10× of frozen retina sections from PEG-Arg1- or PEG-treated *db/db* mice with 3-NT, respectively. (**D**,**F**) Fluorescence intensity quantification of oxidative stress, showing significant reductions in 4-HNE and 3-NT in retinas of PEG-A1-treated *db/db* mice (n = 5–6/group, * *p* < 0.05, ** *p* < 0.01).

**Figure 5 cells-11-02890-f005:**
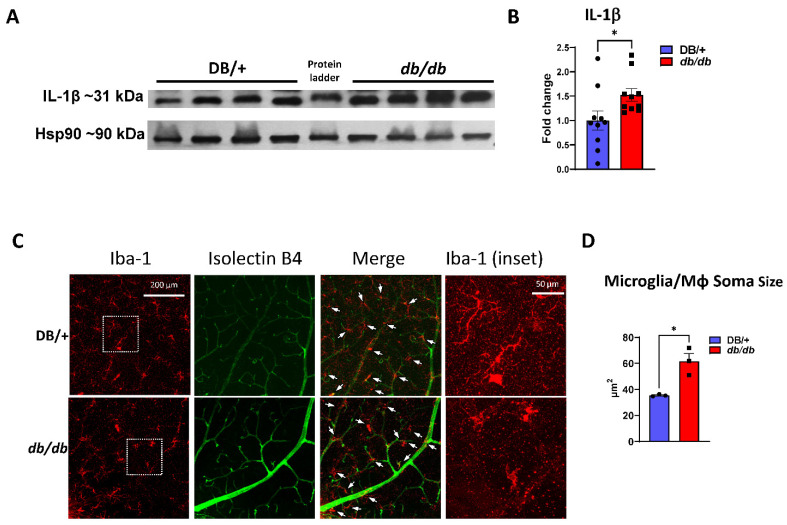
Elevated retinal inflammation and MΦ/microglia soma enlargement in *db/db* mice. (**A**) Representative Western blots of IL-1β in retinal protein lysates with quantification in (**B**), demonstrating significant increases in IL-1β (n = 10) expression in *db/db* mice compared with the lean DB/+ controls. (**C**) Retinal flat-mounts imaged at the level of the superficial vascular plexus and labeled with isolectin B4 (vasculature) and Iba-1 (MΦ/microglia). White arrows point to the perivascular macrophages with Iba-1 in the enlarged inset (boxed in white) showing exceptionally enlarged perivascular MΦ/microglia. (**D**) Quantification of MΦ/microglia soma size, demonstrating a significant size increase in *db/db* mouse retinas compared with the lean controls (n = 3 mice/genotype, * *p* < 0.05). Data are presented as the mean ± SEM.

**Figure 6 cells-11-02890-f006:**
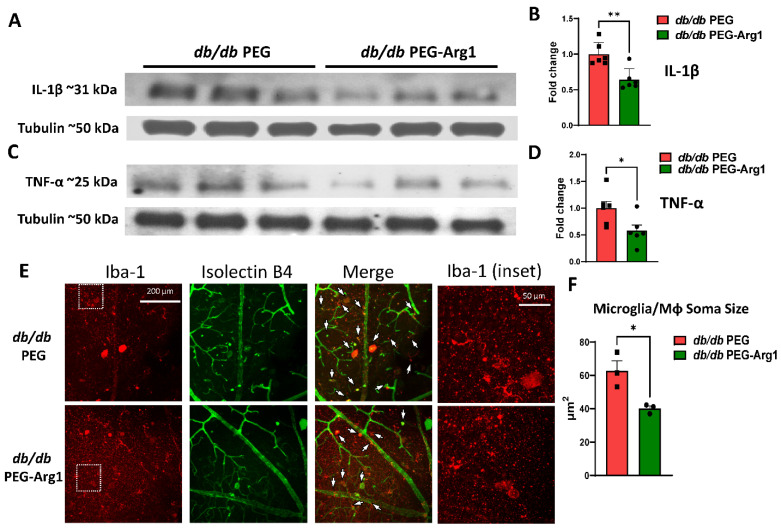
PEG-Arg1 treatment reduced retinal inflammation and reversed MΦ/microglia soma enlargement. (**A**,**C**) Representative Western blots of the pro-inflammatory cytokines IL-1β and TNF- α in retinal protein lysates from PEG-A1-treated *db/db* mice with quantification (**B**,**D**, respectively) compared with the PEG-treated *db/db* controls (n = 6/group, * *p* < 0.05, ** *p* < 0.01). (**E**) Retinal flat-mounts imaged at the level of the superficial vascular plexus labeled with isolectin B4 (vasculature) and Iba-1 (MΦ/microglia) with Iba-1 in the enlarged inset (boxed in white) showing exceptionally enlarged perivascular MΦ/microglia. (**F**) Quantification of MΦ/microglia soma size, demonstrating significant size reductions in the *db/db* mice treated with PEG-Arg1 compared with the vehicle-treated controls (n = 3 mice/group, * *p* < 0.05).

**Figure 7 cells-11-02890-f007:**
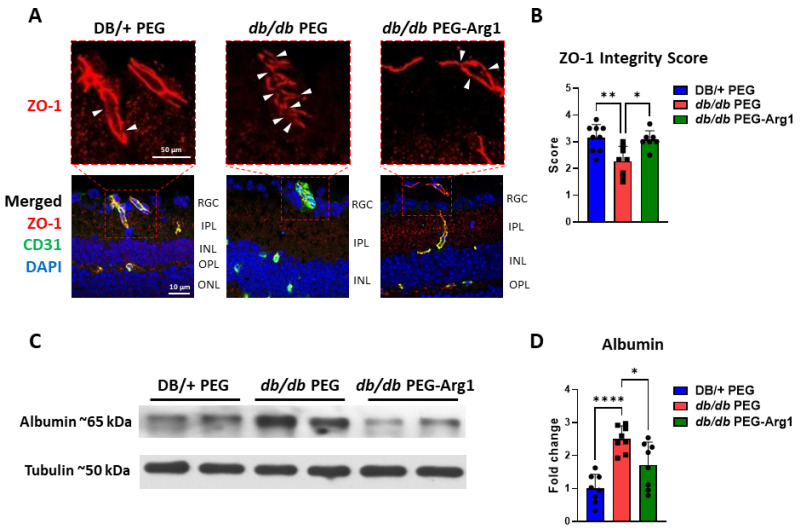
PEG-Arg1 treatment improves the blood–retinal barrier in *db/db* mice. (**A**) Representative immunofluorescent co-labeling of ZO-1 (tight junction protein) and CD31 (endothelial-specific marker) from frozen retina sections from PEG-treated DB/+ (lean control), PEG-treated *db/db* (obese, diabetic control), and PEG-Arg1-treated *db/db* mice, with the enlarged inset (red box) for ZO-1 staining showing tight junction breaks (white triangles). The co-localization of ZO-1 with CD31 demonstrates moderate restoration of the blood–retinal barrier (BRB) in *db/db* mice treated with PEG-Arg1. (**B**) Quantitative analysis of ZO-1 and CD31 immunofluorescent labeling represented as average ZO-1 continuity scores, showing a significant improvement in BRB continuity with PEG-Arg1 treatment. (* *p* < 0.05, ** *p* < 0.01) (**C**) Representative Western blots of retinal tissue albumin extravasation, demonstrating increased albumin extravasation in the PEG-treated control *db/db* mice compared with the lean controls. (**D**) PEG-Arg1 treatment significantly decreased albumin leakage in the retinas of *db/db* mice (n = 8/group, * *p* < 0.05 and **** *p* < 0.001). RGC: retinal ganglion cell layer; IPL: inner plexiform layer; INL: inner nuclear layer; OPL: outer plexiform layer; ONL: outer nuclear layer.

## Data Availability

The data reported here are available from the corresponding author upon reasonable request.

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
