# Peer review of "Systemic Administration of Pegylated Arginase-1 Attenuates the Progression of Diabetic Retinopathy"

_cells, 2022, doi:10.3390/cells11182890_

Round 1

Reviewer 1 Report

The growing number of diabetic patients with limited access to the gold standard 473 

In ‘Systemic administration of pegylated arginase-1 attenuates pro-gression of diabetic retinopathy’, their  results indicate that systemic L-arginine depletion using PEG-Arg1 is effective in reducing inflammation and oxidative stress, restoring BRB integrity, and improving visual acuity in a murine model of diabetic retinopathy. This is a novel finding and an promising method for treating diabetic retinopathy. Minor English check is needed. More animal experiments to prove longterm safety is mandatory before apply to human study.

demonstrated that systemic delivery of PEG-Arg1 can cross the BRB, especially at areas 478 

with pathological barrier dysfunction. Therefore, it could be used without the need for 479 

intravitreal delivery, removing the requirement for highly trained personnel to administer 480 

and reducing the risk of damage to the eye. In addition, PEG-Arg1 therapy has been well- 481 

tolerated in cancer patients, making PEG-Arg1 a promising therapeutic agent for DR, wor- 482 

thy of further investigation and development. 

Author Response

Reviewer 1:

In ‘Systemic administration of pegylated arginase-1 attenuates progression of diabetic retinopathy’, their results indicate that systemic L-arginine depletion using PEG-Arg1 is effective in reducing inflammation and oxidative stress, restoring BRB integrity, and improving visual acuity in a murine model of diabetic retinopathy. Minor English check is needed. More animal experiments to prove longterm safety is mandatory before apply to human study.

We thank and agree with the reviewer for the statement that our study of PEG-Arg1 provides novel findings and a promising method for treating diabetic retinopathy. We also examined our text utilizing a spelling and grammar function of MS Word to check for problems with language and syntax and corrected the very few items uncovered. We did revise the wording of several sentences in the Discussion and Conclusions for clarity. Edits are shown by track changes. Long term safety studies in animals are needed before human use. However, current and past studies of the use of PEG-Arg1 in human cancer treatment trials have provided some proof of safety.

Reviewer 2 Report

 This is a very interesting manuscript that studied the therapeutic effects of PEG-Arg1 in a mouse model of type 2 diabetes. The authors provided convincing data to show that PEG-Arg1 treatment alleviated retinal neuronal dysfunction, inflammation, microglial activation and vascular barrier breakdown in diabetic retinopathy. The discovery is significant and innovative. I only have a few minor suggestions.

1. Please provide an illustration to show the structure of PEG-Arg1. What was the dosage of PEG-5000 used to treat animals?

2. The concentration of PEG-Arg1 in plasma and retina should be provided.

3. For a more precise data presentation, Fig.1 E & F, Y-axis should start at 0.

4. While it is understood that Arg1 will compete with iNOS and inhibit iNOS activity, this study also found that iNOS level was significantly reduced by PEG-Arg1. Please provide some potential mechanisms in the discussion.

Author Response

Reviewer 2:

Thank you for your questions and comments.

1.--Please provide an illustration to show the structure of PEG-Arg1.

No structural illustration is available, but we did describe its structure quantitatively. We have added this in lines (108-110):

“The product of pegylation process attaches 1 to 6 polyethylene glycol 5,000 mw (PEG) units to lysine residues of rhArg1 through the succinamide propionic acid (SPA) linkers; the detailed chemical characterization of PEG-Arg1 is described by Tsui SM et al [23].”.

Dose selection was based on the published literature and our previous studies, and dosing interval was based on the PEG-Arg1 in vivo estimated half-life of 3 days.

2.--What was the dosage of PEG-5000 used to treat animals?

The dose of PEG-5000 was the same as used for PEG-Arg1 (25mg/kg/dose). This dose is described in lines (96-98).

“In the second study, db/db mice between 4-6 months of age were treated with intraperitoneal (IP) injections with either a 25 mg/kg dose of PEG-Arg1 or the same dose and volume of PEG-5000 MW (as vehicle control) thrice weekly for 2 weeks.”

3.--The concentration of PEG-Arg1 in plasma and retina should be provided.

At present, an assay for determining intact PEG-Arg1 concentrations in plasma or tissue is not available. Prior work in our labs using western blot analysis determined a half-life for PEG-Arg1 of approximately 3 days. We modified the text of our results in line (248-254) as follows: “Pharmacokinetic/ pharmacodynamics of PEG-Arg1 has been extensively studied in mice and humans [23, 26]. We have shown previously the bioavailability of PEG-Arg1 in retina after exposure to ischemia/ reperfusion injury due to breakdown of blood-retinal barrier [27]. Using Western blot analysis and immunofluorescence labeling for PEG in frozen retinal sections (Suppl Figure 1A and B), we confirmed the bioavailability of PEG-Arg1 (>250 kD) in the neural retina of db/db mice 3 days after the last IP injection.”

We provided evidence of PEG-Arg1 bioavailability to the retinal tissue in db/db mice in the supplementary figure1?.

4.--For a more precise data presentation, Fig.1 E & F, Y-axis should start at 0.

We agree. The Y-axis for Fig. 1E & 1F and Fig. 2A have been modified.

5.--While it is understood that Arg1 will compete with iNOS and inhibit iNOS activity, this study also found that iNOS level was significantly reduced by PEG-Arg1. Please provide some potential mechanisms in the discussion.

This is an important point. In the Discussion, we do provide evidence from prior study by Lee et al. that ‘expression of iNOS is controlled by its substrate availability, and L-arginine depletion has been shown to reduce iNOS expression (14)’. - Lines 450-451.